# Consideration of Maintenance in Wine Fermentation Modeling

**DOI:** 10.3390/foods11121682

**Published:** 2022-06-08

**Authors:** Alain Rapaport, Robert David, Denis Dochain, Jérôme Harmand, Thibault Nidelet

**Affiliations:** 1MISTEA, Université Montpellier, INRAE, Institut Agro, 34060 Montpellier, France; 2Technord, 7500 Tournai, Belgium; r.david@technord.com; 3ICTEAM, Université Catholique de Louvain, 1348 Louvain-la-Neuve, Belgium; denis.dochain@uclouvain.be; 4LBE, Université Montpellier, INRAE, 11100 Narbonne, France; jerome.harmand@inrae.fr; 5SPO, Université Montpellier, INRAE, Institut Agro, 34060 Montpellier, France; thibault.nidelet@inrae.fr

**Keywords:** wine fermentation, nitrogen, mathematical modeling, population model, maintenance, variable yield

## Abstract

We show that a simple model with a maintenance term can satisfactorily reproduce the simulations of several existing models of wine fermentation from the literature, as well as experimental data. The maintenance describes a consumption of the nitrogen that is not entirely converted into biomass. We show also that considering a maintenance term in the model is equivalent to writing a model with a variable yield that can be estimated from data.

## 1. Introduction

The overall principle of wine fermentation consists of the conversion of sugar into ethanol by yeast. It has been observed for a long time that nitrogen consumed during the yeast growth also plays an important role. The fermentation can be indeed modeled by a two-step process in which the yeast first grows on nitrogen as a limiting resource and then degrades the non-limiting sugar into ethanol and carbon dioxide. However, experimental observations have shown that the consumed nitrogen was not entirely converted into biomass. Several mathematical models were proposed to take these characteristics into consideration. For instance, in [1,2], the biomass growth follows a logistic law whose carrying capacity depends on the initial quantity of nitrogen. In [3], a model that distinguishes part of nitrogen used for yeast growth from another part responsible of the synthesis of proteins (hexose transporters [4]) was developed. Both models were calibrated with different sets of experimental data and provide satisfactory fitting. However, both models present some drawbacks. The dependency of the dynamics on the initial condition of the first model makes it sensitive to the precise knowledge of the initial quantity of nitrogen (that needs to be “memorized” in the dynamical equations of the model). Moreover, it does not allow consideration of non-batch operations or continuous addition of nitrogen, such as in [5] for instance. The second model relies on the knowledge of the time-varying concentration of transporters, which is in general not easily accessible for experimental measurements, and several assumptions were necessary to estimate it from biomass measurements.

The objective of the present work is to propose a new model that reconciles both approaches in a single one.

The observation of the ratio of produced biomass over nitrogen consumption along the whole fermentation, determined on experimental database or numerical simulations of models [1,3], shows that this ratio is non-constant and depends on the initial quantities. This highlights that the conversion of nitrogen into biomass can be viewed as a variable yield process. The experimental evidence that nitrogen is not entirely converted into biomass therefore advocates for the consideration of a maintenance term in the modeling (see, for instance, [6]), without necessarily requiring a detailed representation of the internal mechanism or cells.

Indeed, different mechanisms in the internal functioning of the cells have been investigated in the literature, particularly the role of carbohydrate accumulation [7,8,9], which could explain that the growth dynamics of yeast in wine fermentation does not follow the classical mass-balanced models [10,11]. However, the measurements of these biochemical compounds is experimentally very difficult and is almost impossible in an industrial framework.

The rationale of the results presented here is to test if the introduction of a maintenance term (see [12,13,14] or [15,16,17]) can improve wine fermentation modeling. One of the original features of the proposed approach is to view nitrogen consumption as a global consumption for growth by considering a variable yield. This allows us to avoid to consider a specific structure to model the maintenance. Thus, the purpose of the present work is to investigate the ability of a simpler model with a maintenance term to reproduce and predict wine fermentation kinetics.

Here, we propose a new modeling approach based on a maintenance term (which gives rise to a variable yield), a feature that has not been yet considered in the wine fermentation literature, to the best of our knowledge.

It focuses mainly on the new modeling of the growth of yeast on nitrogen.

This new model was validated using both data generated by existing models (Section 4) and experimental data (Section 5).

## 2. The Proposed Model

We denote by *N*, *S*, *E*, CO${}_{2}$ and *X* the concentrations of (total) nitrogen, sugar, ethanol, dioxide carbon and biomass, respectively. For simplicity, we derive a model under isothermal conditions.

For the first step N→X (yeast growth on nitrogen), we propose the following equations
(1)$$\begin{array}{cc} \frac{dX}{dt}= & \mu_{N}\left(N,X\right)X \end{array}$$
(2)$$\begin{array}{cc} \frac{dN}{dt}= & -\frac{\mu_{N}\left(N,X\right)X}{Y}-m\left(N,X\right)X \end{array}$$
where *Y* is the growth yield, μ the Contois growth function
$$\mu_{N}\left(N,X\right)=\frac{\mu_{N}^{max}N}{N+K_{N}X}$$
and *m* a *maintenance* function, which is positive for N>0 and X>0. We choose here a ratio-dependent kinetics function $\mu_{N}$ to reproduce the observation that the growth is slowing down under an excess of yeast, with a Contois expression as in [3]. In the literature, the maintenance *m* is often considered as constant [12,13], which was validated in continuous culture (chemostat). In general, continuous cultures are intended to be operated at a stationary phase, very differently to batch-operating mode. However, as already investigated in [17], maintenance terms have to depend on the level of available resources; say, *R* (*N* here). In particular, constant maintenance in a batch model would imply $\frac{dR}{dt}<0$ when the resource is exhausted, i.e., R=0, and thus *R* could take unrealistic negative values, as underlined in [14]. In [15,16], the maintenance is directly related to the microbial activity, which is stopped in absence of nutrients. This is why we consider a maintenance function proportional to the growth activity, with a factor that might depend on the nitrogen concentration (one may expect that it decreases when the substrate *N* becomes rare)
$$m\left(N,X\right)=\alpha \left(N\right)\mu_{N}\left(N,X\right)$$
where α is a positive function equal to zero for N=0. Then, one can consider the function *y* defined as follows
$$y\left(N\right):=\frac{Y}{1+\alpha (N)Y},\;N\geq 0$$

Formally, model (Equation 1) and (Equation 2) can be rewritten equivalently as
(3)$$\begin{array}{cc} \frac{dX}{dt}= & \mu_{N}\left(N,X\right)X \end{array}$$
(4)$$\begin{array}{cc} \frac{dN}{dt}= & -\frac{\mu_{N}\left(N,X\right)X}{y(N)} \end{array}$$
where the function *y* is playing the role of a *variable yield*. Identifying the function *m* or the function *y* is thus formally equivalent. However, we shall see in the next section that identifying the function *y* instead of *m* presents some practical advantages.

For the second step $S\to E+{\mathrm{CO}}_{2}$, we follow the model proposed in the literature [3]
(5)$$\begin{array}{cc} \frac{dE}{dt}= & \frac{d{\mathrm{CO}}_{2}}{dt}=\left[\mu_{N}\left(N,X\right)+\beta \nu_{E}\left(E\right)\right]\mu_{S}\left(S\right)X \end{array}$$
(6)$$\begin{array}{cc} \frac{dS}{dt}= & -k\frac{dE}{dt} \end{array}$$
where $\mu_{S}$ is a Monod function and $\nu_{E}$ a function inhibited by the ethanol
(7)$$\mu_{S}\left(S\right)=\frac{\mu_{S}^{max}S}{K_{S}+S},\;\nu_{E}\left(E\right)=\frac{1}{1+K_{E}E}$$

Inhibition by the consumption of sugar *S* by ethanol *E* has been reported many times in the literature [18,19,20,21,22]. The constant yield of production *k* of CO${}_{2}$ and consumption of *S* follows a mass balance assumption, verified experimentally [23], that can be determined using thermodynamics considerations [24].

Note that this model can be extended to anisothermal conditions, considering that the maximal specific rate parameters $\mu_{N}^{max}$, $\mu_{S}^{max}$ and affinity constants $K_{S}$, $K_{E}$ are temperature dependent, as in [3].

## 3. Calibration of the Model

From model Equation (Equation 1), the parameters of the function $\mu_{N}$ can be identified independently of the yield and maintenance terms. To validate the hypothesis of ratio dependency of the function $\mu_{N}$, one can first use experimental data to plot the slope of the logarithm of *X* versus the ratio r=N/X and check if it qualitatively follows a function of the form
$$\mu \left(r\right)=\frac{\mu_{N}^{max}r}{K_{N}+r}$$

A classical least-square method can be applied to fit parameters $\mu_{N}^{max}$, $K_{N}$ on the data. Alternatively, one can plot the inverse of the slope of the logarithm of *X* versus the inverse of the ratio *r* to check if it qualitatively follows a linear dependency, as obtained from Equation (Equation 1)
(8)$${\left(\frac{d\log X}{dt}\right)}^{-1}=\frac{1}{\mu_{N}^{max}}+\frac{K_{N}}{\mu_{N}^{max}}{\left(\frac{N}{X}\right)}^{-1}$$
However, for the accurate identification of the parameters $\mu_{N}^{max}$, $K_{N}$, a linear regression on Equation (Equation 8) is expected to be less reliable than a non-linear least-square optimization of the solution X(·) of (Equation 3), because $\left({\left(\frac{d}{dt}\log X\right)}^{-1},{\left(\frac{N}{X}\right)}^{-1}\right)$ data might be too far to be uniformly distributed.

Note from Equations (Equation 1) and (Equation 2) that one has
$$\lim_{t\to +\infty }N\left(t\right)=0$$
(because the derivative of *N* cannot vanish when *N* is not exhausted). In absence of the maintenance term *m*, one gets $\frac{dX}{dt}+Y\frac{dN}{dt}=0$ which implies that one should have
$$Y=\frac{X(+\infty )-X(0)}{N(0)-N(+\infty )}=\frac{X(+\infty )-X(0)}{N(0)}$$
To test the validity of the model with maintenance, one can plot from experimental data the ratio $\frac{X(+\infty )-X(0)}{N(0)}$ for different values of N(0) to check that it is not constant. If this is the case, one can then look at identifying a non-constant function *y*. For this purpose, we write from Equations (Equation 3) and (Equation 4)
$$X\left(+\infty \right)-X\left(0\right)=-\int_{0}^{+\infty }y\left(N\left(t\right)\right)\frac{dN}{dt}\left(t\right)\;dt$$
and as t↦N(t) is a monotone-decreasing function, one can make the change of variable n=N(t) in this last integral to obtain
$$X\left(+\infty \right)-X\left(0\right)=\int_{0}^{N(0)}y\left(n\right)\;dn$$
Therefore, if one fits a differential function *f* such that f(0)=0 that satisfies
X(+∞)−X(0)=f(N(0))
for experimental data with different values of N(0), then one simply gets $y=f^{\prime }$.

Let us underline that identifying the function *y* in this way can be achieved independently of the knowledge of the kinetics $\mu_{N}$, differently to the function *m*, which clearly presents some robustness advantages. Once the function $\mu_{N}$ is identified, the maintenance function can then be determined as
$$m\left(N,X\right)=\left(\frac{1}{y(N)}-\frac{1}{Y}\right)\mu_{N}\left(N,X\right)$$
where Y=y(0) (to fulfill α(0)=0).

For model Equations (Equation 5) and (Equation 6), the coefficient *k* is kept from the literature, and the parameters β, $\mu_{S}^{max}$, $K_{S}$, $K_{E}$ are identified (with a least-square method) from experimental data of CO${}_{2}$ production rate.

## 4. Validation of the Model on Synthetic Data

We have used synthetic data generated by models of the literature that were previously validated on experimental data [1,3] for a range of initial conditions and operating conditions.

Fitting comparisons of the proposed model with the different data sets are reported in Section 6.

### 4.1. Validation on Simulations of a Model with Transporter

We have considered the model with transporters developed in [3], which is more complex with two additional state variables: the concentrations of hexose transporters and the nitrogen dedicated to these transporters. Data were generated by simulating this model with the parameters given in [3] and operating conditions given in Table 1.

This model explicitly distinguishes two forms of nitrogen, one available for the yeast $N_{X}$ and the other one $N_{tr}$ for the transporters. To compare with the variable *N* of our model, we have considered the total nitrogen $N=N_{X}+N_{tr}$.

#### 4.1.1. Estimation of the Contois Function

We have used a non-linear least-square method based on a Newton algorithm with a finite difference approximation of the Jacobian matrix (function leastsq of scilab). Figure 1 shows a good fitting of the Contois function $\mu_{N}$ on data $\left(\frac{N}{X},\frac{\frac{dX}{dt}}{X}\right)$ of the transporter model, with parameters given in Table 2.

#### 4.1.2. Estimation of the Variable Yield Function

On Figure 2, data X(T)−X(0) versus N(0) from the model with transporters were plotted for T=350 h (we have checked that *N* is quasi-null at *T* and that *X* no longer increases after *T*). One can see that the points are aligned. However, the line that passes through these points does not touch 0, which is not possible for a constant yield (for a constant yield, the points have to be aligned on a line that passes through 0, because when N(0)=0, there is no biomass production).

Then, we fitted a $C^{2}$ function *f* such that f(0)=0 with the following expression
$$f\left(N\right)=\left\{\begin{array}{cc} aN+b\left(1-{\left(\frac{N_{†}-N}{N_{†}}\right)}^{3}\right), & N<N_{†}\\ aN+b, & N\geq N_{†} \end{array}\right.$$
whose parameters are given in Table 3.

The calibration of the parameters *a*, *b* of the function *f* was performed with a linear regression (function reglin of scilab).

Then, we obtain the variable yield function *y* as the $C^{1}$ function
$$y\left(N\right)=f^{\prime }\left(N\right)=\left\{\begin{array}{cc} a+b\frac{3{\left(N_{†}-N\right)}^{2}}{{N_{†}}^{3}}, & N<N_{†}\\ a, & N\geq N_{†} \end{array}\right.$$
and the function α, which describes the maintenance as
$$\alpha \left(N\right)=\frac{1}{y(N)}-\frac{1}{y(0)}=\left\{\begin{array}{cc} \frac{N_{†}^{3}}{aN_{†}^{3}+3b{\left(N_{†}-N\right)}^{3}}-\frac{N_{†}}{3b+aN_{†}}, & N<N_{†}\\ \frac{3b}{a(3b+aN_{†})}, & N\geq N_{†} \end{array}\right.$$
which are both depicted on Figure 3.

Note that the model with transporters was validated only for N(0) in the interval [0.071, 0.57] g·L${}^{-1}$, and that we have no a priori information about the behavior of the yield for values of N(0) smaller than 0.071 g·L${}^{-1}$. The threshold parameter $N_{†}$ was simply chosen so that the simulations of the variables *X* and *N* of the model (Equation 3) and (Equation 4) were the closest to the ones of the transporter model.

#### 4.1.3. Estimation of the Other Parameters and Comparison of the Models

For the model of the second step $S\to E+{\mathrm{CO}}_{2}$, the stoichiometric parameter *k* was taken from the literature, while the other parameters β, $\mu_{S}^{max}$, $K_{S}$, $K_{E}$ were estimated with a least-square optimization on the CO${}_{2}$ chronicles only (the CO${}_{2}$ production rate being a variable that is usually measured in experiments), starting from values in [3]. Values are given in Table 4.

Here, we also used a non-linear least-square method based on a Newton algorithm with a finite difference approximation of the Jacobian matrix (function leastsq of scilab). All data were re-normalized to 1 (i.e., for each variable, the figures were divided by the largest one).

Finally, we present on Figure 4, Figure 5 and Figure 6 simulations of the new model for three largely different initial values of nitrogen from 0.170 g·L${}^{-1}$ to 0.567 g·L${}^{-1}$. The evolution of the ethanol concentration *E* has not been reproduced as it is proportional to the CO${}_{2}$ concentration.

These simulations show the ability of the new model to reproduce, with a single set of parameters, close simulations to the model with transporters, in terms of production of biomass and dioxide carbon, estimation of the peak of the CO${}_{2}$ production rate and depletion of (total) nitrogen and sugar.

### 4.2. Validation on the SOFA Model

The model proposed in [1] does not explicitly consider transporters with an additional state variable as the previous model, and instead presents a more sophisticated expression of the dynamics that depend on the initial condition, with an additional latency term at the beginning of the simulations.

Differently to the previous model, which is built as a “mass-balanced” model, this one relies on an empirical dynamics of logistic shape for the biomass growth, with some parameters that depend on the initial concentration of nitrogen N(0), instead of the two-dimensional model (Equation 3) and (Equation 4).

Therefore, this is not a Markovian model. It has been validated on different operating conditions, and has been encoded into the SOFA software exploited for decision-making [2]. We launched simulations of this model for the same operating conditions than for the previous model (Table 1). Although simulations look qualitatively similar, they do not overlap, especially for the biomass chronicle. This could be explained by the fact that this model is intended to predict a number of cells and not a precise biomass (an average number of $4.15\times 10^{9}$ cells for one *g* of biomass was used to have *X* expressed in g·L${}^{-1}$ as for the previous model). We proceeded to a new validation of our model on these data.

#### 4.2.1. Estimation of the Contois Function

Figure 7 shows that the data $\left(\frac{N}{X},\frac{\frac{dX}{dt}}{X}\right)$ do not precisely follow the graph of a function (this is most probably due to the fact that the model is not Markovian). Indeed, this happens mainly for the large value $N_{0}$ of the initial nitrogen. We believe that this could be explained by the dynamics of the biomass *X* of this model, which is a logistic law with a carrying capacity given by an heuristic expression that depends on $N_{0}$, and not dynamics coupled with the dynamics of *N* (indeed the interval of tested values of $N_{0}$ might be larger than the validity of this model). However, we have fitted the graph of a Contois function to these data with the parameters given in Table 5, which was able to satisfactorily reproduce the trajectories of the model for a large amplitude of values of $N_{0}$, as we shall see later on.

As for the previous model, we used a non-linear least-square method based on a Newton algorithm with a finite difference approximation of the Jacobian matrix (function leastsq of scilab). As one can see in Table 5, the values of $\mu_{N}^{max}$ and $K_{N}$ are significantly larger and smaller, respectively, than in Table 2, which is consistent with the observation that this model predicts a faster convergence of the biomass to its maximal value, despite the latency term (compare Figure 4, Figure 5 and Figure 6 with Figure 8, Figure 9 and Figure 10).

#### 4.2.2. Estimation of the Variable Yield Function

Data X(T)−X(0) from the simulation of the SOFA model were plotted on Figure 11 at T=350 h, for different values of N(0) in the interval [0.071, 0.57] g·L${}^{-1}$ (here, we also checked that the fermentation was quasi-ended at *T*). One can see that the points follow an increasing concave curve and further increase very slowly, quite differently to the model with transporters (see Figure 2).

We have then fitted a $C^{2}$ function *f* with f(0)=0 for the expression
$$f\left(N\right)=\left\{\begin{array}{cc} bN-aN^{2}, & N<N_{†}\\ bN-aN^{2}+bN+\frac{A}{B}\left(e^{-BN_{†}}-e^{-BN}\right) & N<N_{†} \end{array}\right.$$
with
$$A=\left(b-2aN_{†}\right)e^{BN_{†}},\;B=\frac{2a}{b-2aN_{†}}$$
and parameters *a*, *b*, $N_{†}$ given in Table 6.

Parameters *a* and *b* were determined with a linear regression (function reglin of scilab).

Then, we obtain the expression of the variable yield function
$$y\left(N\right)=f^{\prime }\left(N\right)=\left\{\begin{array}{cc} b-2aN, & N<N_{†}\\ Ae^{-BN}, & N\geq N_{†} \end{array}\right.$$
as well as the function α
$$\alpha \left(N\right)=\frac{1}{y(N)}-\frac{1}{y(0)}=\left\{\begin{array}{cc} \frac{1}{b-2aN}-\frac{1}{b}, & N<N_{†}\\ \frac{e^{b(N-N_{†})}}{b-2aN_{†}}-\frac{1}{b}, & N\geq N_{†} \end{array}\right.$$
whose graphs are drawn on Figure 12.

#### 4.2.3. Estimation of the Other Parameters and Comparison of the Models

For the second step, the same stochiometric parameter *k* was taken for the literature, and the other parameters β, $\mu_{S}^{max}$, $K_{S}$, $K_{E}$ were estimated with a least-square optimization on the CO${}_{2}$ chronicles only, as for data generated by the model with transporters (see Table 7).

Figure 8, Figure 9 and Figure 10 show the comparison between the SOFA model and our calibrated model for the same initial condition than for the former comparison with the model with transporters. Here also, we see that the proposed model reproduces quite faithfully the simulations of the SOFA model, with the advantage of being a simpler Markovian model. Indeed, the difference between the model with transporters and the SOFA model can be translated into different maintenance terms (see Figure 3 and Figure 12): for large values of nitrogen, the model with transporters behaves like a model with a maintenance proportional to the growth, while the SOFA model amounts to have a strongly increasing maintenance. Recall that the simulations for the largest value of N(0) showed the most differences between these two models (for N(0)=0.567 g·L${}^{-1}$, the model with transporters predicts a biomass production of 5.11 g·L${}^{-1}$, while the SOFA model predicts 3.88 g·L${}^{-1}$; see Figure 6 and Figure 10). While the model with transporters was validated experimentally for N(0) in the interval [0.170, 0.567] g·L${}^{-1}$, we believe the validation of the SOFA model for initial concentrations of nitrogen larger than 0.4 g·L${}^{-1}$ might need to be revisited (although our model once calibrated is able to reproduce the SOFA simulations).

## 5. Calibration of the Model on Real Data

We considered data from experiments conducted at SPO Lab (INRAE, Montpellier, France) in 2004, that were used to calibrate the model with transporters and the SOFA model (see [1,3]). The data consisted of a set of three experiments with the same operating conditions given in Table 1 and different initial concentrations N(0) of nitrogen, exactly the same as for the simulations of Section 4.1 and Section 4.2. For each experiment, one had

-Height measurement points for *X*.-No measurement point for *N*, *S* or *E*.-About 400 measurement points for CO${}_{2}$ and $d{\mathrm{CO}}_{2}/dt$.

We first calibrated a function f(·) to the data (N(0),X(T)−X(0)), with the same expression as in Section 4.2, to determine a yield function y(·) (see Figure 13), using a linear regression to estimate parameters *a* and *b*.

As we do not have measurements of *N* over the time, we cannot estimate the Contois parameters independently of the CO${}_{2}$ measurements, as we did with the synthetic data. All the parameters of the model were fitted simultaneously with a least-square method (values are given in Table 8), except for the sugar-conversion yield, for which we have used the value of the literature k=2.17, as before.

The non-linear least-square method uses a Newton algorithm with a finite difference approximation of the Jacobian matrix (function leastsq of scilab), and the data set was re-normalized to the maximal value of 1.

Figure 14, Figure 15 and Figure 16 show the results of the fitting for the three experiments. One can appreciate the goodness of fit for a unique set of parameters. In particular, the production of biomass and CO${}_{2}$, as well as the height and date of the peak of $d{\mathrm{CO}}_{2}/dt$, are well predicted with this model.

## 6. Fitting Comparisons

For the calibration of the variable yield function on both synthetic and experimental data (Section 4 and Section 5), we have used a linear regression (function reglin of scilab) for the determination of parameters *a*, *b* of the function *f* (for the model with transporter) and $f^{\prime }$ (for the SOFA model and experimental data). The residual error is given in Table 9.

This shows that the model with transporters behave very closely to a variable yield model. The fitting performances for the SOFA model and experimental data are more difficult to interpret, because the validity of the SOFA model for the large range of initial concentrations of nitrogen we considered is questionable, and the quantity of experimental data is quite poor compared to the synthetic data.

For the synthetic data, the calibration of the growth characteristics (parameters $\mu_{N}^{max}$, $K_{N}$ of the Contois function) was performed first, independently of the CO${}_{2}$ data. Then, parameters for the second step (parameters *k*, β, $\mu_{S}^{max}$, $K_{S}$, $K_{E}$ for the CO${}_{2}$ production) were calibrated. In both cases, a non-linear least-square method based on a Newton algorithm with a finite difference approximation of the Jacobian matrix (function leastsq of scilab) was used. Table 10 shows a good fitting quality.

We recall that for experimental data, we do not have measurement of *N* over time, so it was not possible to estimate the growth function independently of the CO${}_{2}$ measurements. The estimation of all the parameters was made on the CO${}_{2}$ measurements only. We have used the same non-linear least-square method, with data re-normalized to 1 (i.e., the figures were divided by the largest one), so that all points have equal weight in the criterion. The errors shows a good fitting of the CO${}_{2}$ curves with the model with maintenance.

## 7. Conclusions

In this work, we demonstrated that the consideration of a maintenance term, or equivalently, a variable yield, in wine fermenting modeling can satisfactorily replace more sophisticated models with a simpler structure. Indeed, the effects of the underlying mechanisms of transporters or carbohydrate accumulation, which are difficult to capture experimentally, are somehow encoded into a maintenance term, and are translated into a variable yield between biomass and nitrogen. We showed that this variable yield, as a function of the nitrogen concentration, can be estimated from experimental data of biomass growth and nitrogen depletion, without the need to measure internal compounds. This consideration brings a flexibility to suit different kind of models or experimental data (once calibrated) with a single common structure, that could correspond to different operating conditions or hypotheses in wine fermentation. This new approach provides new perspectives of control of fermentation with nitrogen addition, based on a simple Markovian model, as well as model extensions with aromatic compounds [25] or multi-strains [26]. 

## Figures and Tables

**Figure 1 foods-11-01682-f001:**
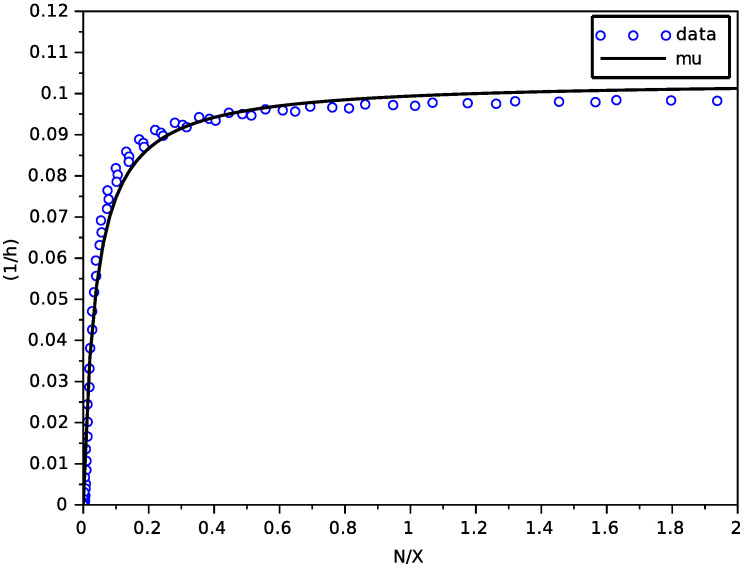
Result of the fitting of the Contois function on data from the model with transporters.

**Figure 2 foods-11-01682-f002:**
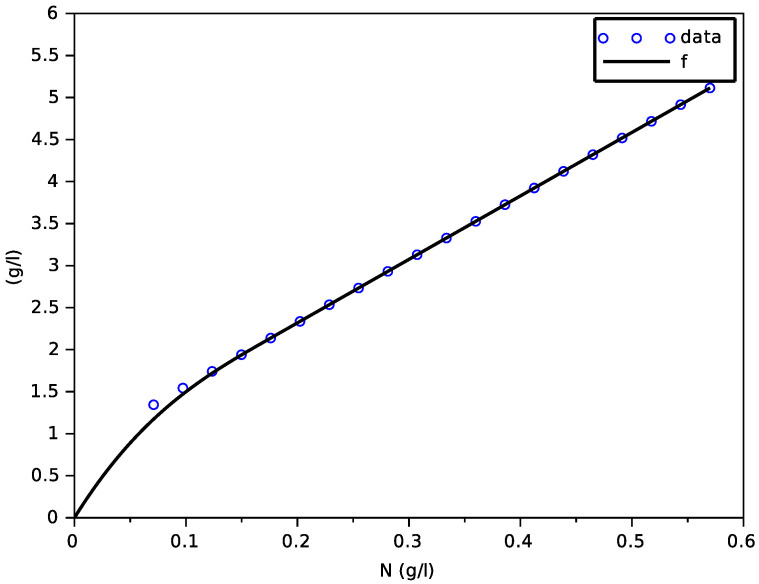
Result of the fitting of the function *f* on data from the model with transporters.

**Figure 3 foods-11-01682-f003:**
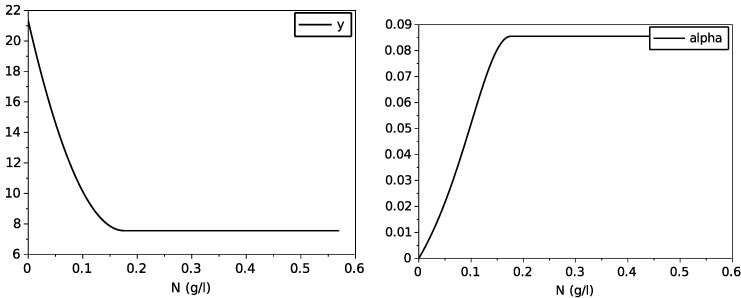
Graphs of the obtained variable yield function *y* and of the function α.

**Figure 4 foods-11-01682-f004:**
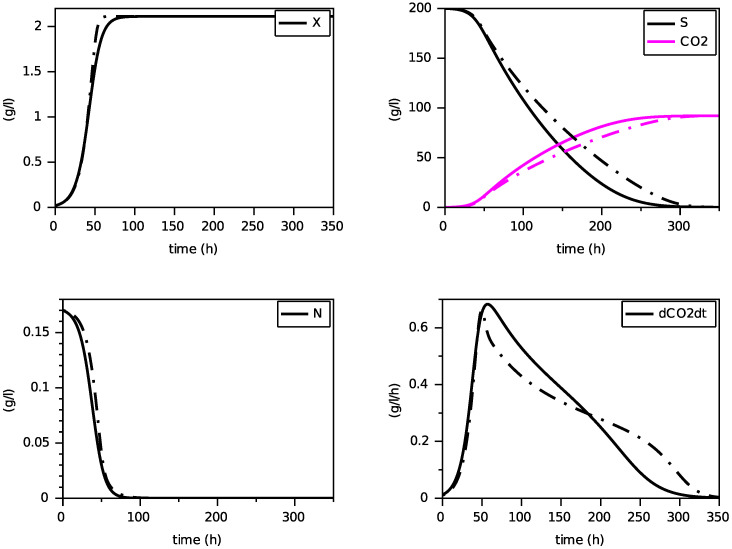
Comparison with the model with transporters (in dashed) for N(0)=0.170 g·L${}^{-1}$ (constant temperature of $24{\;}^{\circ }$C).

**Figure 5 foods-11-01682-f005:**
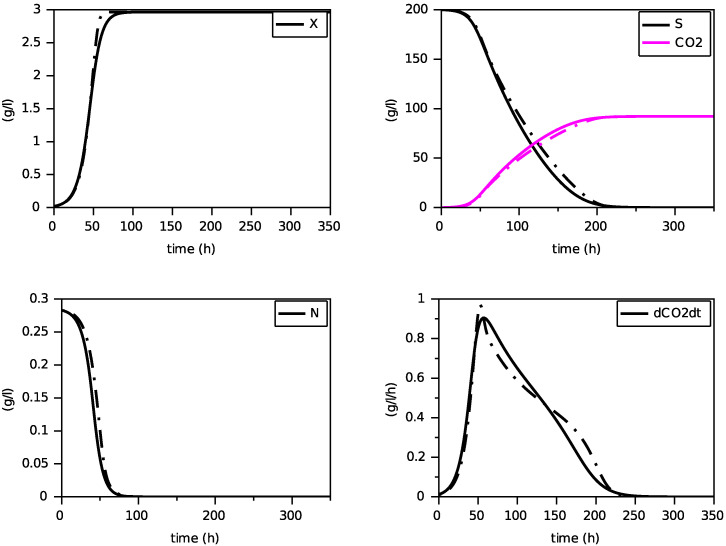
Comparison with the model with transporters (in dashed) for N(0)=0.283 g·L${}^{-1}$ (constant temperature of $24{\;}^{\circ }$C).

**Figure 6 foods-11-01682-f006:**
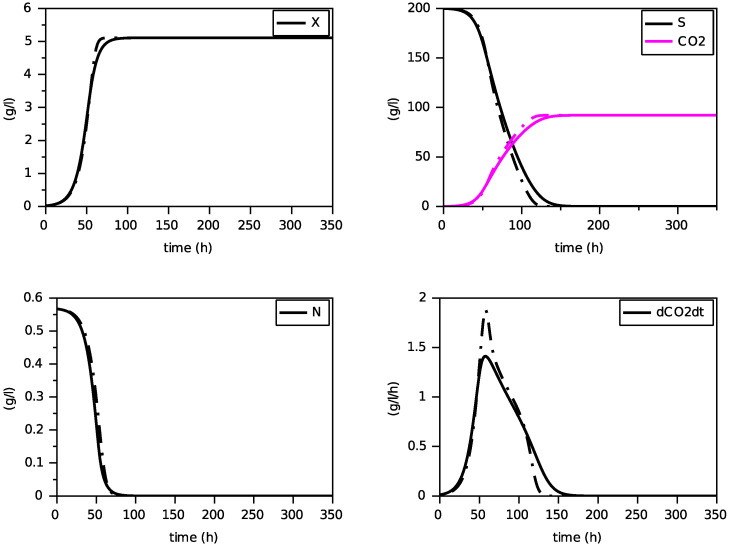
Comparison with the model with transporters (in dashed) for N(0)=0.567 g·L${}^{-1}$ (constant temperature of $24{\;}^{\circ }$C).

**Figure 7 foods-11-01682-f007:**
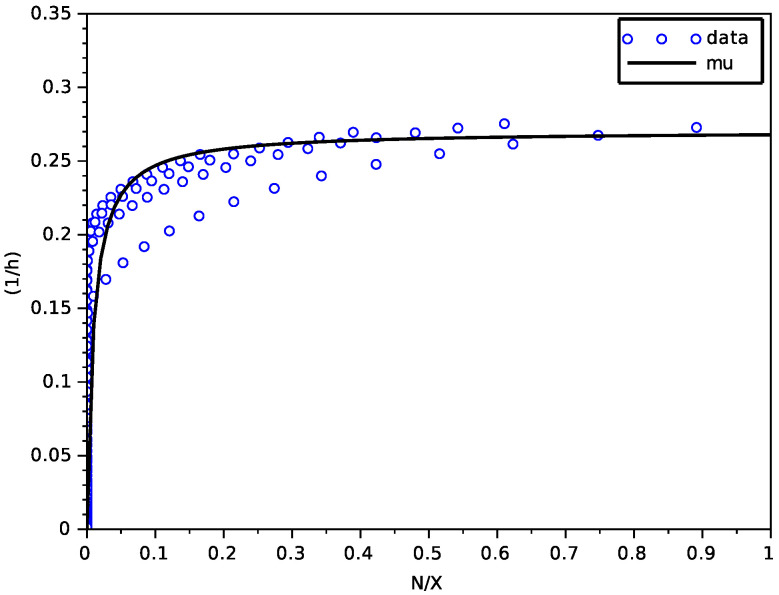
Result of the fitting of the Contois function on data from the SOFA model.

**Figure 8 foods-11-01682-f008:**
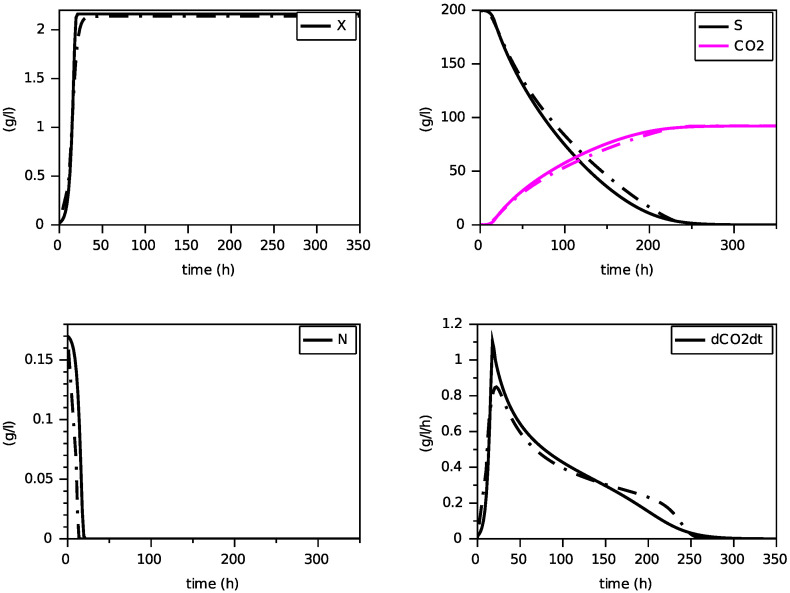
Comparison with the SOFA model (in dashed) for N(0)=0.170 g·L${}^{-1}$ (constant temperature of $24{\;}^{\circ }$C).

**Figure 9 foods-11-01682-f009:**
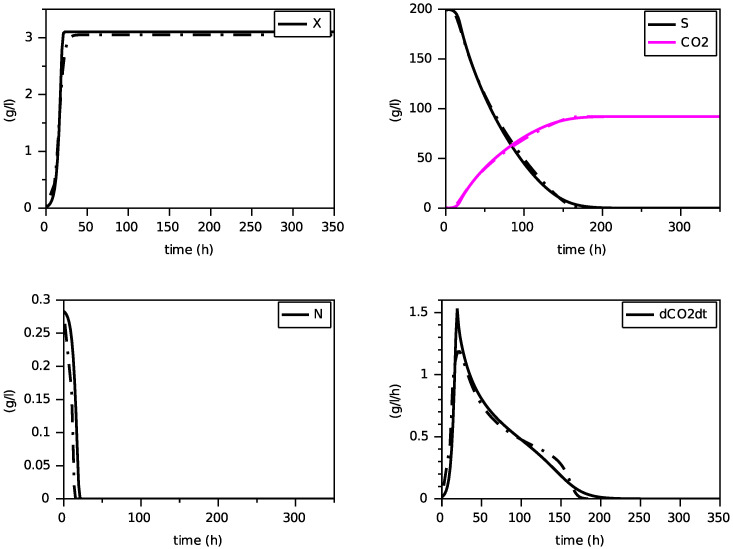
Comparison with the SOFA model (in dashed) for N(0)=0.283 g·L${}^{-1}$ (constant temperature of $24{\;}^{\circ }$C).

**Figure 10 foods-11-01682-f010:**
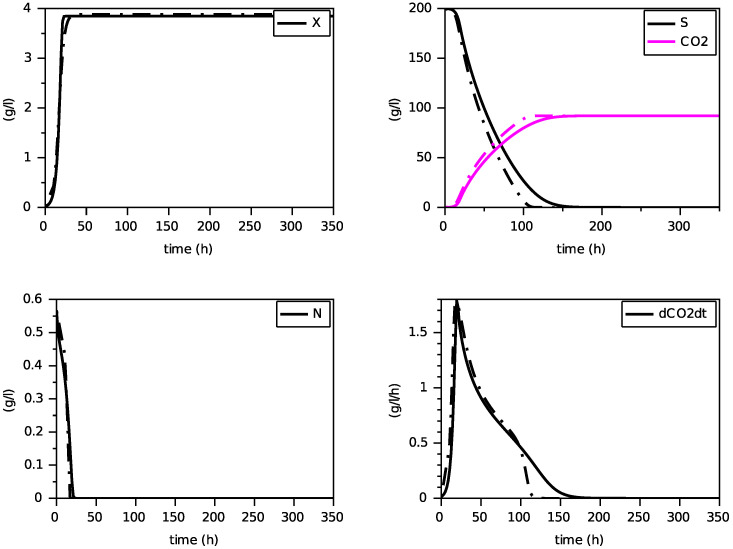
Comparison with the SOFA model (in dashed) for N(0)=0.567 g·L${}^{-1}$ (constant temperature of $24{\;}^{\circ }$C).

**Figure 11 foods-11-01682-f011:**
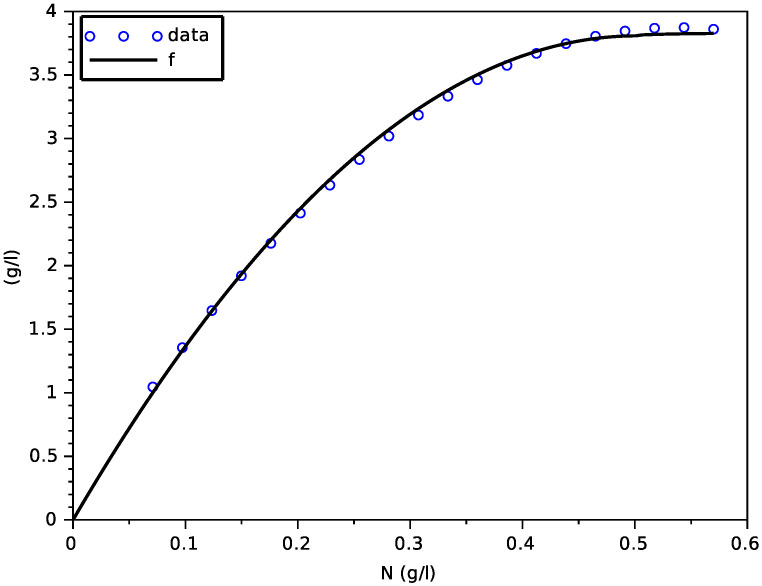
Result of the fitting of the function *f* on data from the SOFA model.

**Figure 12 foods-11-01682-f012:**
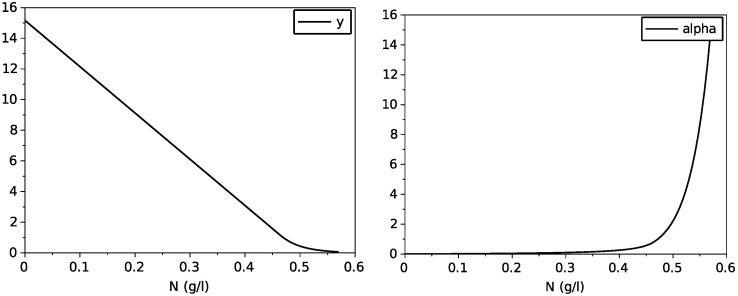
Graphs of the obtained variable yield function *y* and of the function α.

**Figure 13 foods-11-01682-f013:**
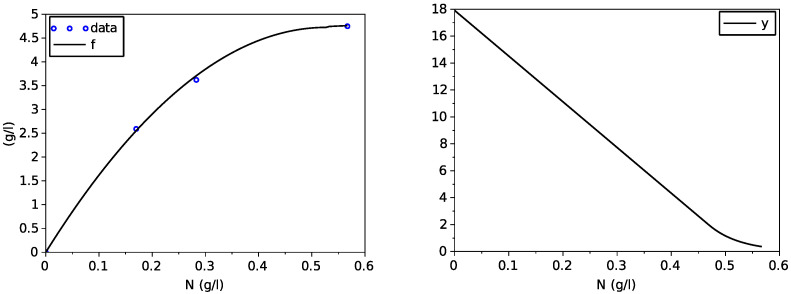
Results of the fitting of the function *f* on the experimental data (**left**) and of the corresponding variable yield function *y* (**right**).

**Figure 14 foods-11-01682-f014:**
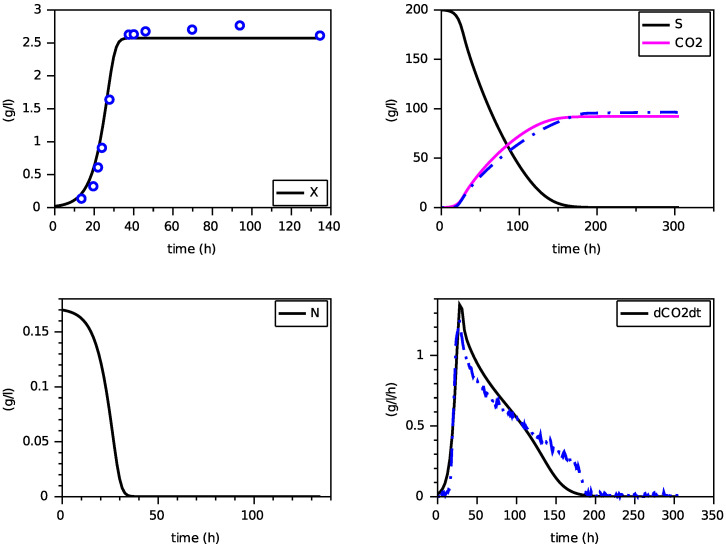
Simulation for N(0)=0.170 g·L${}^{-1}$ (experimental data in blue).

**Figure 15 foods-11-01682-f015:**
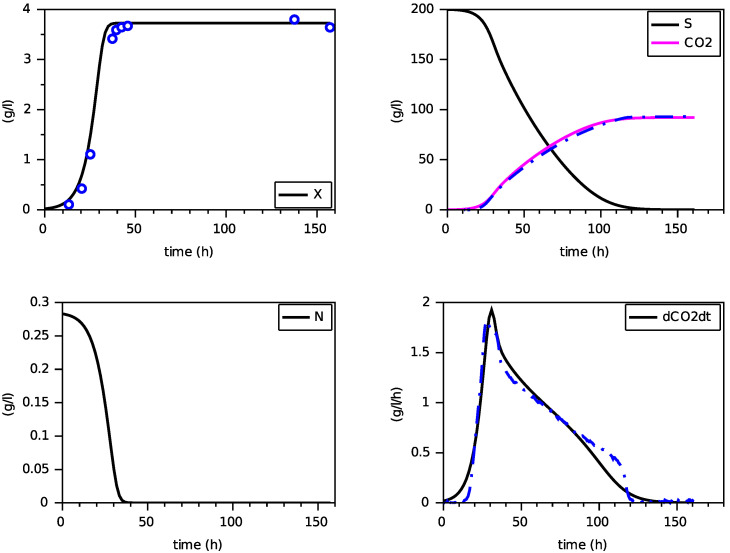
Simulation for N(0)=0.283 g·L${}^{-1}$ (experimental data in blue).

**Figure 16 foods-11-01682-f016:**
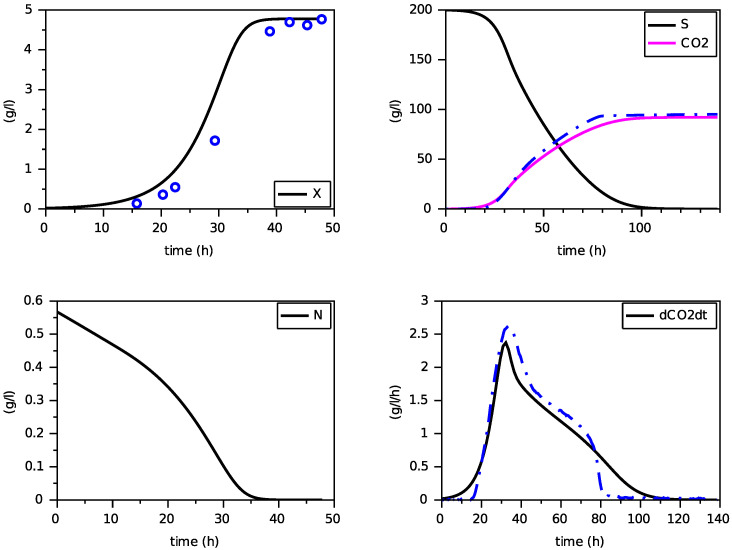
Simulation for N(0)=0.567 g·L${}^{-1}$ (experimental data in blue).

**Table 1 foods-11-01682-t001:** Operating conditions for the simulation of the model with transporters.

X(0)	0.02 g·L${}^{-1}$
N(0)	0.071–0.57 g·L${}^{-1}$
S(0)	200 g·L${}^{-1}$
time horizon	350 h
temperature	constant equal to $24^{\circ }$
others	no initial transporter
	no nitrogen addition

**Table 2 foods-11-01682-t002:** Parameters of the Contois function $\mu_{N}$.

$\mu_{N}^{max}$	0.103 h${}^{-1}$
$K_{N}$	0.0381 g·L${}^{-1}$

**Table 3 foods-11-01682-t003:** Parameters of the variable yield function *y*.

*a*	7.55
*b*	0.808 g·L${}^{-1}$
$N_{†}$	0.176 g·L${}^{-1}$

**Table 4 foods-11-01682-t004:** Parameters for the second step $S\to E+{\mathrm{CO}}_{2}$ model.

*k*	2.17
β	2.41
$\mu_{S}^{max}$	0.197 h${}^{-1}$
$K_{S}$	21.1 g·L${}^{-1}$
$K_{E}$	72.7 g·L${}^{-1}$

**Table 5 foods-11-01682-t005:** Parameters of the Contois function $\mu_{N}$.

$\mu_{N}^{max}$	0.270 h${}^{-1}$
$K_{N}$	0.00952 g·L${}^{-1}$

**Table 6 foods-11-01682-t006:** Parameters of the variable yield function *y*.

*a*	15.1 g·L${}^{-1}$
*b*	15.2
$N_{†}$	0.465 g·L${}^{-1}$

**Table 7 foods-11-01682-t007:** Parameters for the second step $S\to E+{\mathrm{CO}}_{2}$ model.

*k*	2.17
β	3.22
$\mu_{S}^{max}$	0.197 h${}^{-1}$
$K_{S}$	17.6 g·L${}^{-1}$
$K_{E}$	36.4 g·L${}^{-1}$

**Table 8 foods-11-01682-t008:** Parameters fitted on the experimental data.

$\mu_{N}^{max}$	0.175 h${}^{-1}$
$K_{N}$	0.0133 g·L${}^{-1}$
β	1.622
$\mu_{S}^{max}$	0.393 h${}^{-1}$
$K_{S}$	19.2 g·L${}^{-1}$
$K_{E}$	71.9 g·L${}^{-1}$

**Table 9 foods-11-01682-t009:** Residual standard error (RSE) for the determination of *a* and *b*.

Data	Tr Model	SOFA Model	Exp.
RSE	$2.21\times 10^{-10}$	0.199	0.225

**Table 10 foods-11-01682-t010:** Root Mean Square Error (RMSE) for the calibration of the growth function μ and the CO${}_{2}$ chronicles.

Data	Tr Model	SOFA Model	Exp.
RMSE (μ)	0.0414	0.292	-
RMSE (CO${}_{2}$)	0.0543	0.0895	0.0519

## Data Availability

The data presented in this study are available on request from the corresponding author.

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
