# Peer review of "Consideration of Maintenance in Wine Fermentation Modeling"

_foods, 2022, doi:10.3390/foods11121682_

Round 1

Reviewer 1 Report

The objective of this study was to develop a mathematical model with a maintenance term can satisfactorily reproduce the simulations of several existing models of wine fermentation from the literature.

This is an interesting, extensive research, with a lot of numerical analysis. The experimental results were takken from the literature!
Thematically the work could attract intention of the researchers and professionals. Also, the proposed manuscript is relevant to the scope of the journal.

The overall organization and structure of the manuscript seem to be appropriate. The paper is well written and the topic is appropriate for the journal.

The aim of the paper is well described and the discussion was well approached, its results and discussion are correlated to the cited literature data.

The literature review is comprehensive and properly done.

In the final paragraphs of the introductory section the authors explain what is the core of their research. However, it has to be clearly stated by the authors what is their contribution that makes the research different enough in comparison to the other authors' works and they have to further elaborate the extent of novelty in their research.

The novelty of the work must be more clearly demonstrated.

The significance of the Work: Given the large number of analyzed data, this is an interesting study with a possible significant impact in this area.

Perhaps a more detail  description of SOFA model should be shown, regarding the initial values, bounding conditions, and the obtained results. Statistical interpretation of the analytical data must be more properly presented. The verification of the model should be performed. Model validation is possibly the most important step in the model building sequence. 

Other Specific Comments: The work is properly presented in terms of the language. The work presented here is very interesting and well done, it is presented in a compact manner.

The methodology applied in the research is presented in clear manner, so that it is repeatable by other authors.

The main drawback of the paper could be the extent of novelty, or the main novelty in the present work, compared to the works of other researchers? In my opinion, the authors should put additional effort to demonstrate that the present work gives a substantial contribution in the research area.

The manuscript should be improved from technical/graphical viewpoint. The graphs should be plotted in the similar form, some of them are outlined, some are larger then others, the font size seem to be different, etc.

Author Response

The authors thank the Reviewer for its careful reading and his relevant comments and suggestions. The modifications appear in blue color in the revised version. We give below our answers to the points raised by the Reviewer.

In the final paragraphs of the introductory section the authors explain what is the core of their research. However, it has to be clearly stated by the authors what is their contribution that makes the research different enough in comparison to the other authors' works and they have to further elaborate the extent of novelty in their research.

The novelty of the work must be more clearly demonstrated.

Answer. The consideration of maintenance term in wine fermentation modelling is new, up to our knowledge. Moreover, we show in this work that one does not have to impose an a priori structure of the maintenance term, the difficulty having been circumvented by the identification of a “variable yield” on data. Existing literature that proposes to consider additional compartments (transporters, carbohydrate accumulation…) in the model are usually not identifiable in practice because these additional variables are not accessible to measurements, which is not the case of the proposed model. Several sentences have been added in blue in the introduction and the conclusion to emphasis these facts.

Perhaps a more detail description of SOFA model should be shown, regarding the initial values, bounding conditions, and the obtained results.

Answer. To better describe this model, we have added the following sentence: “Differently to the former model which is built as a “mass-balanced” model, this one relies on an empirical dynamics of logistic shape for the biomass growth, with some parameters that depend on the initial concentration of nitrogen N(0), instead of the two dimensional model (3)-(4).” The results of this model are also presented on Figures 10, 11, 12.

Statistical interpretation of the analytical data must be more properly presented. The verification of the model should be performed. Model validation is possibly the most important step in the model building sequence.

Answer. Additional information about the fitting methods for the validation have been added. A new section 6 presents the statistical results of the calibrations.

The main drawback of the paper could be the extent of novelty, or the main novelty in the present work, compared to the works of other researchers? In my opinion, the authors should put additional effort to demonstrate that the present work gives a substantial contribution in the research area.

Answer. As underlined above, we have added at several places in the introduction what appear to us as novelties in this work.

The graphs should be plotted in the similar form, some of them are outlined, some are larger than others, the font size seem to be different, etc.

Answer. All graphs have been redraw, unified and completed with units.

Reviewer 2 Report

The authors provide an interesting model of wine fermentation that advocates for including a maintenance term associated with nitrogen limitation. From a physiological point of view, I think the usage of the term maintenance might be confusing. Some other terminology might better reflect the variable effect yield mentioned in this paper. 

In addition, I think the quality of this work could be significantly improved if the authors considered the actual experimental data from David et al. (2013) instead of using simulations from a previous model. 

In my opinion, it would be essential to include some relevant missing literature. A possible explanation for the experimental observations would be that yeast accumulates carbohydrates when nitrogen becomes scarce. It could be pertinent to discuss the maintenance hypothesis against the carbohydrate accumulation hypothesis. Additional literature:

Explores the role of carbohydrate accumulation: Vargas, Felipe A., et al. "Expanding a dynamic flux balance model of yeast fermentation to genome-scale." BMC systems biology 5.1 (2011): 1-12.

Shows carbohydrate accumulation after nitrogen depletion: Schulze, Ulrik, et al. "Physiological effects of nitrogen starvation in an anaerobic batch culture of Saccharomyces cerevisiae." Microbiology 142.8 (1996): 2299-2310.

Shows carbohydrate accumulation after nitrogen depletion. Varela, Cristian, Francisco Pizarro, and Eduardo Agosin. "Biomass content governs fermentation rate in nitrogen-deficient wine musts." Applied and environmental microbiology 70.6 (2004): 3392-3400.

Uses a dynamic biomass equation: Henriques, David, et al. "A multiphase multiobjective dynamic genome-scale model shows different redox balancing among yeast species of the saccharomyces genus in fermentation." Msystems 6.4 (2021): e00260-21.

Models carbohydrate accumulation with experimental data from Varela and Schulze: Henriques, David, and Eva Balsa-Canto. "The Monod Model Is Insufficient To Explain Biomass Growth in Nitrogen-Limited Yeast Fermentation." Applied and Environmental Microbiology 87.20 (2021): e01084-21.

------------------------------------------------------------------------------------

Some additional comments:

Below, I think some clarification is needed. I will try to break it into propositions to explain why:

Line 63: "The observation of the ratio of produced biomass over nitrogen consumption along the whole fermentation, determined on experimental database or numerical simulations of models [6, 13], shows that this ratio is non-constant and depends on the initial quantities. The experimental evidence that nitrogen is not entirely used for growth therefore advocates for the consideration of a maintenance term in the modeling (see for 44 instance [10])."

A=This highlights that the conversion of nitrogen into biomass can be viewed as a variable yield process.  

B=The experimental evidence that nitrogen is not entirely used for growth 

-Does A imply B or is there some experimental data to support B ? I tend to agree with claim A, but I doubt B. Other factors such as carbohydrate accumulation could explain A without the need for B. Double-check reference 10; I think it refers to a software/model paper. 

-Check reference 13? 

Line 63: "In the literature, the maintenance m is often considered as constant [15, 16], which has been validated in continuous culture (chemostat)."

Does this refer to the NGAM in the stoichiometric models? In my view, this makes sense for energy (i.e, carbon limited chemostat). The maintenance of cellular processes requires ATP, which requires a fermentable substrate. On the other hand, in the case of nitrogen, autophagy allows some recycling of nitrogen. Can the authors elaborate more on this or provide additional references?

Equation 2: I think there is a typo... should read dN/dt instead of dX/dt.

-----------------------------------------------------------------------------

General:

Add units to figures.

Check units in variable description.

Elaborate on figure captions.

Author Response

The authors thank the Reviewer for its careful reading and his relevant comments and suggestions. The modifications appear in blue color in the revised version. We give below our answers to the points raised by the Reviewer.

From a physiological point of view, I think the usage of the term maintenance might be confusing. Some other terminology might better reflect the variable effect yield mentioned in this paper.

Answer. We feel a bit embarrassed by this remark. Indeed, we have followed the development proposed originally by S. Pirt, which is highly cited, that explicitly defines a “maintenance” term. By maintenance, we understood a generic expression that gathers all the possible underlying mechanisms that use substrate or energy not for the direct conversion into biomass growth. We are prrone to change this expression but we do not see what other term could be less confusing than maintenance… Any suggestion from the Reviewer is welcome.

I think the quality of this work could be significantly improved if the authors considered the actual experimental data from David et al. (2013) instead of using simulations from a previous model.

Answer. Indeed, this is exactly what we did in Section 5. The interest of a comparison with simulations of existing models (Section 4) instead of real data is that we can show that our model can reproduce accurately the trajectories of all the state variables, even the ones for which we have few experimental data (in particular, the nitrogen concentration was measured at initial time but not online…)

In my opinion, it would be essential to include some relevant missing literature. A possible explanation for the experimental observations would be that yeast accumulates carbohydrates when nitrogen becomes scarce. It could be pertinent to discuss the maintenance hypothesis against the carbohydrate accumulation hypothesis. Additional literature…

Answer. We thank the Reviewer to point out this literature. This possible explanation is in line with the transporter hypothesis, and concerns internal bio-chemical compounds which are not easily accessible to practical measurements… Our approach consists indeed of considering a generic “variable yield” that can be identified from experimental measurements, independently of any assumption or interpretation of the underlying mechanisms of the maintenance. Of course we agree that it does provide a thorough model for understanding. Additional references and a paragraph mentioning this explanation have been added in the introduction.

Line 63: "The observation of the ratio of produced biomass over nitrogen consumption along the whole fermentation, determined on experimental database or numerical simulations of models [6, 13], shows that this ratio is non-constant and depends on the initial quantities. The experimental evidence that nitrogen is not entirely used for growth therefore advocates for the consideration of a maintenance term in the modeling (see for 44 instance [10])."

A=This highlights that the conversion of nitrogen into biomass can be viewed as a variable yield process. 

B=The experimental evidence that nitrogen is not entirely used for growth

Does A imply B or is there some experimental data to support B ? I tend to agree with claim A, but I doubt B. Other factors such as carbohydrate accumulation could explain A without the need for B.

Answer. We believe there was probably a confusion (of our fault) between “not entirely used for growth” and “not entirely converted into biomass”. We agree that growth cannot be reduced to a simple biomass augmentation. Therefore, we have replaced in the text “not entirely used for growth” by “not entirely converted into biomass”.

Double-check reference 10; I think it refers to a software/model paper. Check reference 13?

Answer. Ref. 10 describes both a model and its companion software. An accent was missing in Ref. 13.

Line 63: "In the literature, the maintenance m is often considered as constant [15, 16], which has been validated in continuous culture (chemostat)."

Does this refer to the NGAM in the stoichiometric models? In my view, this makes sense for energy (i.e, carbon limited chemostat). The maintenance of cellular processes requires ATP, which requires a fermentable substrate. On the other hand, in the case of nitrogen, autophagy allows some recycling of nitrogen. Can the authors elaborate more on this or provide additional references?

Answer. We agree with the Reviewer that relating maintenance with energy considerations is certainly relevant, and this has been already addressed in the literature, since the pioneer work of Pirt. However, our objective in this work is precisely to show that it is not necessary to enter into these considerations to obtain and calibrate a model that could satisfactorily reproduce and predict wine fermentation at a population (or macroscopic) level. Concerning the chemostat, we wanted to underline that for continuous cultures operating most of the time at stationary state, a constant maintenance term is acceptable, but not in batch as we consider here, because it would violate the non-positivity of the state variable. A remark in this sense has been added in the text.

Equation 2: I think there is a typo... should read dN/dt instead of dX/dt.

Answer. This has been corrected: thank you!

Add units to figures. Check units in variable description. Elaborate on figure captions.

Answer. All the figures have been revisited and improved.